# OpenReview forum: "SafetyPairs: Isolating Safety Critical Image Features with Counterfactual Image Generation"
_ICLR.cc/2026/Conference — Submitted to ICLR 2026_

### Official Review · Reviewer_4LMv · 2025-10-25

**Soundness:** 2
**Presentation:** 2
**Contribution:** 2
**Rating:** 2
**Confidence:** 3

**Summary:**

This paper proposes SAFETYPAIRS, a scalable framework that generates counterfactual image pairs differing only in safety-relevant features. Using instruction-based image editing and automated consistency checks with LLMs, the authors construct a dataset of 3,020 images (1,510 pairs) across 9 safety categories. The benchmark allows for systematic evaluation of the guardrail model’s sensitivity to subtle safety cues and can also serve as effective data augmentation for lightweight guard models.

**Strengths:**

1. The paper introduces an automated counterfactual image generation pipeline that isolates safety-critical visual features via targeted image editing.
2. The paper presents a new benchmark dataset (SAFETYPAIRS) that captures fine-grained safe/unsafe distinctions, which can be a valuable resource for future research.

**Weaknesses:**

1. The edit success rate is relatively low (23%), which limits scalability and makes it difficult to expand the dataset for large-scale training purposes. Could the authors try more advanced models, like Nano banana, or do more preprocessing before editing the image to improve the success rate?
2. The data source is too narrow—SafetyPairs is entirely built upon the LlavaGuard dataset. The authors should consider incorporating additional unsafe datasets (e.g., Unsafebench) to improve data diversity and make the benchmark more comprehensive and challenging.
3. The paper does not evaluate any guardrail-specific models, such as LlavaGuard, LlamaGuard3, or classifier-based guard models like Q16 or NudeNet. This omission is strange given that SafetyPairs is derived from LlavaGuard.
4. The training discussion is brief, and the experiments are limited to training a simple linear probe model, which reduces the practical impact of the work.

**Questions:**

1. During evaluation, the authors only consider a binary classification setup. Could the authors provide more detailed metrics or insights into multi-class performance?
2. Have the authors tried to fine-tune VLMs or classifier-based models using the SafetyPairs dataset to assess potential performance improvements?
3. How do the authors handle ambiguous or borderline cases where it is difficult to determine whether an image is safe or unsafe—are such cases manually labeled by multiple labelers or filtered out?

---

> ### Author Response · Authors · 2025-11-21
>
> We thank the reviewer for their time and their review.
> First, we would like to highlight the strengths highlighted by various reviewers (also see general response):
>
> 1. SafetyPairs is an **automated pipeline for generating image pairs that isolate safety-critical visual features** (hn4i, i781, 4LMv) and a **novel benchmark dataset** (9SNW, i781, 4LMv)
> 2. SafetyPairs **effectively highlights prevalent weaknesses in current VLMs** (hn4i, 9SNW, i781)
> 3. SafetyPairs **will prove useful to the community when made publicly available** (9SNW, 4LMv)
> 4. SafetyPairs **easily builds on top of established safety taxonomies and datasets** and **is easily reproducible** (9SNW)
> 5. SafetyPairs augmentations **demonstrate improvements in training safeguard models** (hn4i, 9SNW)
> 6. Our analysis provides **interesting details on failure modes and correlations with image pairs similarity** (9SNW)
>
> ---
>
> We would like to answer the reviewer's specific questions.
>
> > 1. During evaluation, the authors only consider a binary classification setup. Could the authors provide more detailed metrics or insights into multi-class performance?
>
> This is a good suggestion. We added a more thorough breakdown of the performance of our model across different safety categories in the Appendix.
>
> > 2. Have the authors tried to fine-tune VLMs or classifier-based models using the SafetyPairs dataset to assess potential performance improvements?
>
> In this paper, we primarily focused on creating a scalable synthetic data generation pipeline for evaluating the weaknesses of models, and we did not show larger scale training-based approaches beyond our CLIP experiments.
>
> > 3. How do the authors handle ambiguous or borderline cases where it is difficult to determine whether an image is safe or unsafe—are such cases manually labeled by multiple labelers or filtered out?
>
> To address this, we provided detailed annotation and guidelines and examples of edge cases in the Appendix D of the updated manuscript. Due to the sensitive nature of this kind of safety data, the annotation was done by the authors. Examples that seemed ambiguous or borderline in the context of each given policy were filtered out.
>
> ---
>
> We would also like to address some of the reviewers other concerns:
>
> > The edit success rate is relatively low (23%), which limits scalability
>
> We would like to clarify, the $23\\%$ figure is not the edit success rate, but the error rate of the VQA check --- thus our method is scalable as also echoed by the other reviewers (hn4i).
>
> > Could the authors try more advanced models [for image editing]
>
> The reason we can not use a closed-source model like Nano Banana is because we need to edit harmful images, and the safety guardrail models of closed-source APIs restricts us from doing this. This is why we chose the open-source Flux Kontext model which we can run locally.
>
>
> ---
>
> Thanks again for your feedback! If our responses and new results are satisfactory, we would greatly appreciate the reviewer increasing their score to reflect their increased confidence in our work.

---

### Official Review · Reviewer_i781 · 2025-10-27

**Soundness:** 2
**Presentation:** 3
**Contribution:** 2
**Rating:** 2
**Confidence:** 4

**Summary:**

This paper mainly constructs a benchmark dataset that consists of image pairs differing in harmful features. Specifically, the authors leverage image editing models to modify safety-related attributes, making the images safe while keeping other safety-irrelevant aspects unchanged. By using this new safety benchmark, the paper sheds light on the weakness of vision-language models in distinguishing between subtly different images. The benchmark dataset can also serve as a data augmentation resource.

**Strengths:**

- The idea is interesting that SAFETYPAIRS contains a pair of images that differ only in safety-related details. So that it can help clearly see if the VLMs understand the safety-related features.
- New dataset.
- Shed light on the weaknesses of the current VLMs.

**Weaknesses:**

- In the data construction, unsafe images are real-world samples, while safe images are synthetically generated through editing. This introduces a potential confounding variable: a classifier trained on such data might learn to distinguish between real and synthetic images rather than between unsafe and safe content. For example, in Section 4.3, the authors train linear probe models. Could the authors analyze to disentangle these two effects and demonstrate that the model is not just learning a real vs. fake classifier?

- The discussion on related work could be significantly improved. The paper should provide a more detailed comparison with recent works like LlavaGuard and UnsafeBench (Qu et al., 2025). A deeper analysis of the datasets is needed. What are the key advantages of the proposed image pair structure compared to the datasets in LlavaGuard and UnsafeBench? A more thorough discussion would better highlight the unique contributions of this work. The authors should also discuss the guardrail models proposed in those papers. Even including them as baselines to provide a more comprehensive assessment. There appears to be a factual error. The paper claims it contains "entirely synthetically generated images," but the UnsafeBench paper states it includes both real-world and AI-generated images.

- Another concern lies in the evaluation of generalization. The authors train the linear probe model on the SafetyPairs dataset, which is constructed from edited samples of the LlavaGuard dataset, and then evaluate it again on LlavaGuard. This setup means that both the training and evaluation data come from highly similar distributions. As a result, the reported improvements may only reflect in-distribution performance, rather than demonstrating true out-of-distribution generalization. To convincingly support the claimed generalization ability, the method should be evaluated on a dataset that the model has never encountered during training.

- Overall, while the proposed benchmark effectively reveals the nuanced weaknesses of current vision-language models, it falls short of providing a clear roadmap for how this dataset can be leveraged to substantially advance the state of the art in guardrail models.

**Questions:**

- Could the authors analyze to disentangle these two effects and demonstrate that the model is not just learning a real vs. fake classifier?
- What are the key advantages of the proposed image pair structure compared to the datasets in LlavaGuard and UnsafeBench?
- What about the out-of-distribution generalization?
- What are the roadmaps for pushing forward current guardrail models?

---

> ### Author Response · Authors · 2025-11-21
>
> We thank the reviewer for their time and their review.
> First, we would like to highlight the strengths highlighted by various reviewers (also see general response):
>
> 1. SafetyPairs is an **automated pipeline for generating image pairs that isolate safety-critical visual features** (hn4i, i781, 4LMv) and a **novel benchmark dataset** (9SNW, i781, 4LMv)
> 2. SafetyPairs **effectively highlights prevalent weaknesses in current VLMs** (hn4i, 9SNW, i781)
> 3. SafetyPairs **will prove useful to the community when made publicly available** (9SNW, 4LMv)
> 4. SafetyPairs **easily builds on top of established safety taxonomies and datasets** and **is easily reproducible** (9SNW)
> 5. SafetyPairs augmentations **demonstrate improvements in training safeguard models** (hn4i, 9SNW)
> 6. Our analysis provides **interesting details on failure modes and correlations with image pairs similarity** (9SNW)
>
> ---
>
> First we would like to clarify some of the reviewer's concerns:
>
> **We clarify some confusing wording in our related works section, and add a more extensive discussion of related image safety datasets.**
>
> > The paper claims it contains "entirely synthetically generated images," but the UnsafeBench paper states it includes both real-world and AI-generated images.
>
> From our related works section we stated: *"Motivated by the cost of collecting large scale safety datasets, recent work incorporates AI generated images (Qu et al., 2025). However, entirely synthetically generated images run the risk of not covering the same image distribution as real-world unsafe examples."*
>
> We would like to clarify that we are not saying that the UnsafeBench dataset (Qu et al., 2025) is "entirely synthetically generated", but that we avoided using individual images that were entirely synthetically generated, as opposed to real images with synthetic edits.
>
> We clarified our wording and added more discussion of the other related works.
>
> ---
>
> We hope to answer the reviewer's questions below:
>
> > 1. Could the authors analyze to disentangle these two effects and demonstrate that the model is not just learning a real vs. fake classifier?
>
> Thanks for asking. We are confident our model is not learning a real vs fake classifier, because **we evaluate our approach on a withheld test set of LlavaGuard images which are not synthetic**.
>
> > 2. What are the key advantages of the proposed image pair structure compared to the datasets in LlavaGuard and UnsafeBench?
>
> The primary advantage that we highlight is that our paired synthetic images prove more challenging than the out of the box LlavaGuard dataset. Furthermore, our paired structure proves an effective source of data for training lightweight guard models.
>
> > 3. What about the out-of-distribution generalization?
>
> We evaluated our approach on withheld samples from our SafetyPairs evaluation set and the LlavaGuard evaluation set. These datasets both cover a diverse taxonomy of 9 categories, and capture a variety of different safety policies. Expanding our dataset and evaluation to very specific notions of harm, and expanding to different cultures would be a great future direction.
>
> > 4. What are the roadmaps for pushing forward current guardrail models?
>
> In our experiments, we found that there is a strong negative association between the cosine similarity of an image pair in the encoder's representation and a VLMs ability to correctly classify a pair of images (see Figure 8). This indicates that mitigating problems with visual encoders could remedy this safety vulnerability, which we agree is a fruitful avenue for future work. This is consistent with other findings from the literature (see Tong et al., 2024).
>
> ---
>
> Thanks again for your feedback! If our responses and new results are satisfactory, we would greatly appreciate the reviewer increasing their score to reflect their increased confidence in our work.

---

> > ### Comment · Reviewer_i781 · 2025-11-28
> > **Response to authors**
> >
> > Thanks for the clarification! It partially addresses my concerns, but some remain. (1) If I understand correctly, the benchmark dataset was built based on the LlavaGuard dataset. Can these withheld samples truly be considered out-of-distribution? (2) The vulnerability is obvious, yet I do not see a clear path for how current guardrail models could improve.

---

### Official Review · Reviewer_9SNW · 2025-11-01

**Soundness:** 3
**Presentation:** 3
**Contribution:** 3
**Rating:** 6
**Confidence:** 5

**Summary:**

This work introduces safetypairs, a dataset of counterfactual image for safety assessment consisting of 1510 image pairs.
The authors also propose a general data generation framework that can be utilized to generate counterfactual image pairs for safety testing. Given an unsafe image the safetypairs setup leverages VLMs and image editing models to generate a safe counterfactual that remains as close as possible to the original image. This works experiments demonstrate that the SafetyPairs benchmarks remains challenging for models that perform well on non-counterfactual evaluations. Additionally, the authors provide insights into failure mode of VLMs and demonstrate correlations with pair similarity.

**Strengths:**

- counterfactual pairs for safety assessments mark a valuable contribution that unearth prevalent failure modes in safeguard models despite saturation of existing benchmarks

- assuming that the benchmark and code will be made publicly available these will prove useful to the community
- easily builds on top of established safety taxonomies and datasets (here LlavaGuard)
- the data creation framework is easily reproducible. The overall setup is described well and relevant prompts for reproduction are shared.
- demonstrates meaningful improvements in training safeguards on SafetyPair augmentations
- analysis provides some interesting details on failure modes and correlations with image pair similarity

**Weaknesses:**

# Major

- The paper lacks details on the human verification of edit attempted edit pairs and only states that that this was conducted by the authors. While this is generally acceptable (especially in safety related research), at least the Appendix should include some details on the specific setup of this validation, measurements taken to ensure no confounding or bias by the authors and usual demographic statistics on the involved annotators.

- Creating synthetic counterfactuals through generative image editing may lead to a distribution shift of the outputted images [1,2]. The paper provides no control experiment to ensure that potential gaps in qualification accuracy for example are due to a mismatch in real vs synthetic images. Additionally, when using this dataset in training there is a clear confounder with all unsafe image being real and all safe ones synthetically manipulated which can serve as a potential confounder leading to Clever Hans behavior in the classifier

- The evaluation only considers zero-shot VLMs but no dedicated image safety models (LlavaGuard, SHIELDAGENT [3], OpenAI moderation API [4], etc.)

- at the same time one demonstrated benefit of SafetyPairs is their benefit on training/tuning models but these results are lacking. While the small-scale analysis on classification heads is nice the paper would benefit from applying this training to an actual VLM-based guardrail that was also the focus in the evaluation.
Further, since the method constructs pairs DPO would be an obvious application that the paper does not explore currently.

-The paper should provide a datasheet for the newly introduced dataset (https://arxiv.org/abs/1803.09010) to adhere with standardized documentation practices, especially in safety related fields.

## Minor Comments

- paper is limited to one source taxonomy and unsafe dataset (i.e. LlavaGuard). While the methodology should be easily transferrable to other setups the paper would benefit from results along those lines

- the Appendix could be extended with further qualitative samples (ideally some randomly drawn once) to also provide better coverage across models and safety categories

- building safety classifiers w/ linear heads on CLIP representations was first proposed by Q16 and should be cited in Sec. 4.3 [5] (https://arxiv.org/abs/2202.06675)

[1] https://www.semanticscholar.org/paper/Convergence-Dynamics-and-Stabilization-Strategies-Gao-Li/9330b7a20bd171e1a418518c18a21c80630fd8f9

[2] https://www.semanticscholar.org/paper/PRISM%3A-Precision-Recall-Informed-Data-Free-via-He-Wang/83a8776e8f6f8aa711357c4362e166882c8368c2

[3] https://arxiv.org/pdf/2503.22738

[4] https://platform.openai.com/docs/guides/moderation

 [5] https://arxiv.org/abs/2202.06675

**Questions:**

- Q1) can the authors provide further details on their analyses by types of unsafely? For example considering the used LLavaGuard taxonomy are there certain categories that models perform significantly better or worse in? How do more fine-grained performance evaluations translate to observations on pair similarity and ROC tradeoff?

- Q2) How do counterfactuals effect models trained w/o them? For example consider LlavaGuard models. Given that they were trained on the unsafe version of these images, do models exhibit a stronger bias in missclaifiyng the safe counterfactuals?

- Q3) Do you have any insights on further scaling counterfactual data curation and using it in large scale training?

- Q4) Do you have any results indicating if SafetyPairs can be used for paired post-training like DPO? Is there an expected mode collapse here w/ one side of the pair always being synthetic?

---

> ### Author Response · Authors · 2025-11-21
>
> We thank the reviewer for their time and their review.
> First, we would like to highlight the strengths highlighted by various reviewers (also see general response):
>
> 1. SafetyPairs is an **automated pipeline for generating image pairs that isolate safety-critical visual features** (hn4i, i781, 4LMv) and a **novel benchmark dataset** (9SNW, i781, 4LMv)
> 2. SafetyPairs **effectively highlights prevalent weaknesses in current VLMs** (hn4i, 9SNW, i781)
> 3. SafetyPairs **will prove useful to the community when made publicly available** (9SNW, 4LMv)
> 4. SafetyPairs **easily builds on top of established safety taxonomies and datasets** and **is easily reproducible** (9SNW)
> 5. SafetyPairs augmentations **demonstrate improvements in training safeguard models** (hn4i, 9SNW)
> 6. Our analysis provides **interesting details on failure modes and correlations with image pairs similarity** (9SNW)
>
> ---
>
> We would like to answer the reviewers questions and address their concerns.
>
> > The paper lacks details on the human verification ...
>
> This is a very valid concern. **We updated the manuscript with extensive details about our human verification procedure in Appendix D.** As the reviewer mentioned, due to the safety related nature of this work, we (the authors) validated the data ourselves rather than through crowdsourcing. We provided the specific annotation guidelines that we followed when verifying the data, as well as demographic information about the authors.
>
> > Can the authors provide further details on their analyses by types of unsafely? For example considering the used LLavaGuard taxonomy are there certain categories that models perform significantly better or worse in? How do more fine-grained performance evaluations translate to observations on pair similarity and ROC tradeoff?
>
> This is a great question. We added a breakdown of each model by safety category in the Appendix.
>
> > Do you have any insights on further scaling counterfactual data curation and using it in large scale training?
>
> Good question, while our experiments primarily focused on evaluating the weakness of models, we do show evidence in our training-based experiments that training with **unfiltered augmented examples** can improve guard model generalization. Because we don't do human validation on these example this makes our approach for synthetic data generation quite scalable.
>
> > Do you have any results indicating if SafetyPairs can be used for paired post-training like DPO?
>
> To our understanding, a dataset for performing DPO on a VLM **would contain pairs of text responses, rather than pairs of images (what SafetyPairs provides).** So it is not immediately clear how we would leverage SafetyPairs for this type of training setup.
>
> ---
>
> We also would like to clarify some other concerns.
>
> > The paper provides no control experiment to ensure that potential gaps in qualification accuracy for example are due to a mismatch in real vs synthetic images.
>
> **The way we avoided this potential confound was by evaluating on withheld LlavaGuard images which are non-synthetic.** For our training-based experiments shown in Figure 11, we trained on our partially synthetic augmented pairs but did not use these for our evaluation to avoid this exact scenario.
>
> > The paper should provide a datasheet for the newly introduced dataset
>
> This is a great suggestion. We documented our dataset according to this standard in Appendix E of the updated manuscript.
>
> ---
>
> Thanks again for your feedback! If our responses and new results are satisfactory, we would greatly appreciate the reviewer increasing their score to reflect their increased confidence in our work.

---

### Official Review · Reviewer_hn4i · 2025-11-05

**Soundness:** 3
**Presentation:** 3
**Contribution:** 2
**Rating:** 6
**Confidence:** 3

**Summary:**

This work proposes to identify visual features that contribute to "unsafe" images by synthesizing counterfactual edits. Specifically, the authors start from a dataset of unsafe images and prompt LLM to generate edit instructions, such that the edited image no longer violates the safety policy. They use an instruction-based image editing model to perform the edits, and a VQA model to verify the output images.
The counterfactual image pairs are used to evaluate LVLMs on detecting unsafe images. Experimental results suggest that this benchmark is more challenging than those using natural images like LlavaGuard, and that most LVLMs struggle to differentiate between safety-critical features between images. Training with counterfactual editing as data augmentation is also found to improve sample efficiency.

**Strengths:**

- The paper presents a promising data pipeline to perform counterfactual edits on safety features in images. The method appears to be scalable and yields realistic examples that serve as hard negatives for unsafe image detection.
- Experimental results on various VLMs of varying size, both open-source and proprietary. The generated data appears universally challenging for all models tested.
- Even linear probing with a small number of counterfactual samples (<32) can significantly improve guard model performance.

**Weaknesses:**

- I feel that the work has yet to realize the full potential of counterfactual probing. I would have liked to see a more in-depth experimental analysis, providing insights into questions such as
  - What features mattered the most to ground-truth image safety (by analyzing the edited regions/objects? even obtaining the ROIs for unsafe images would be informative),
  - What features the guard models are the most sensitive to (maybe repeat the data pipeline but only change unimportant features?), or
  - Whether certain spurious correlation exists to mislead the models to consider the counterfactual images as unsafe (even after removing the true unsafe features).

  I believe understanding these problems will be helpful for building guard models with higher precision and minimize unwanted false positives from biased data.

- The data pipeline involves human in the loop for 1) verifying image edits and 2) annotating unsafe images w/ rationale (as in LlavaGuard), both of which crucial for high-quality benchmarking data but limits scaling to larger-scale training (e.g., if one were to incorporate a significant amount of safety data into VLM post-training). It would be interesting to explore whether training with more unverified counterfactual data can also improve guard performance, relying on the automated filtering by the VQA models.

**Questions:**

I am curious if the original captions $c_p$ are necessary for generating edit instructions, given the GPT 4o model can understand the images natively. Have the authors done any preliminary experiments comparing the edit quality to a simpler pipeline without the captioning step?

---

> ### Author Response · Authors · 2025-11-21
>
> We thank the reviewer for their time and their review.
> First, we would like to highlight the strengths highlighted by various reviewers (also see general response):
>
> 1. SafetyPairs is an **automated pipeline for generating image pairs that isolate safety-critical visual features** (hn4i, i781, 4LMv) and a **novel benchmark dataset** (9SNW, i781, 4LMv)
> 2. SafetyPairs **effectively highlights prevalent weaknesses in current VLMs** (hn4i, 9SNW, i781)
> 3. SafetyPairs **will prove useful to the community when made publicly available** (9SNW, 4LMv)
> 4. SafetyPairs **easily builds on top of established safety taxonomies and datasets** and **is easily reproducible** (9SNW)
> 5. SafetyPairs augmentations **demonstrate improvements in training safeguard models** (hn4i, 9SNW)
> 6. Our analysis provides **interesting details on failure modes and correlations with image pairs similarity** (9SNW)
>
> ---
>
> We would like to address the reviewer's question:
>
> > Have the authors done any preliminary experiments comparing the edit quality to a simpler pipeline without the captioning step?
>
> This is a good question, we think that it is highly possible that a model could directly generate edit instructions without captioning. The primary reasons for captioning the image are (a) because it is used for collecting atomic facts in teh source image for our VQA constraints and (b) because it comes at little additional cost.
>
> In addition to captioning images, we used human written rationales sourced from the LlavaGuard dataset instead of synthetically generated rationales because we want our dataset to specifically target scenarios that VLMs are likely to overlook. This would mean  relying on VLMs to detect the harmful aspects of images to edit would be circular.
>
> ---
>
> We would also like to address the reviewers concerns:
>
> > The data pipeline involves human in the loop ... limits scaling to larger-scale training
>
> We would like to clarify that **for our training experiments we only used pairs that were not filtered by humans.** This is something that we clarify in the updated manuscript, thank you for pointing this out.
>
> > [Human in the loop is] crucial for high-quality benchmarking
>
> We agree with the reviewer, that human validation is necessary for high quality benchmarking which was our motivation for doing a rigorous human validation phase.
>
> In an effort to more thoroughly document our human validation process, and at the request of reviewer 9SNW, we have also added details about our human validation procedure in Appendix D.
>
> ---
>
> Thanks again for your feedback! If our responses and new results are satisfactory, we would greatly appreciate the reviewer increasing their score to reflect their increased confidence in our work.

---

### Author Response · Authors · 2025-11-21

We thank the reviewers for their thorough responses. We are glad the reviewers highlight some of the strengths of our work:

1. SafetyPairs is an **automated pipeline for generating image pairs that isolate safety-critical visual features** (hn4i, i781, 4LMv) and a **novel benchmark dataset** (9SNW, i781, 4LMv)
2. SafetyPairs **effectively highlights prevalent weaknesses in current VLMs** (hn4i, 9SNW, i781)
3. SafetyPairs **will prove useful to the community when made publicly available** (9SNW, 4LMv)
4. SafetyPairs **easily builds on top of established safety taxonomies and datasets** and **is easily reproducible** (9SNW)
5. SafetyPairs augmentations **demonstrate improvements in training safeguard models** (hn4i, 9SNW)
6. Our analysis provides **interesting details on failure modes and correlations with image pairs similarity** (9SNW)

We have resolved these primary concerns shared by multiple reviewers:
1. **We clarify that we do not require human in the loop for generating training data (hn4i, 9SNW).**  While we use human validation when creating our evaluation benchmark, we do **not** in our training experiments, indicating the potential scalability of our approach.
2. **We clarify that there is no overfitting to real vs synthetic image differences, (9SNW, i781)**  because in our experiments we evaluate our models on the LlavaGuard evaluation set which is **entirely non-synthetic**.
3. **We added the standard datasheet documentation to the Appendix E (9SNW).**
4. **We provided a breakdown of the performance of various models on the different sub-categories of SafetyPairs in Appendix F (4LMv).**

---

### Meta-Review · Area_Chair_1i57 · 2025-12-25

**Summary:**

The submission presents "SafetyPairs," a framework and benchmark for generating counterfactual image pairs to isolate safety-critical features in Vision Language Models (VLMs). The reviewers generally appreciated the automated pipeline and the potential utility of the dataset in highlighting model vulnerabilities. However, opinions were sharply divided. While two reviewers found the contribution sufficient for acceptance, the other two strongly recommended rejection. The critical criticisms focused on the work's heavy reliance on the single LlavaGuard dataset, the omission of key state-of-the-art guardrail baselines (e.g., LlamaGuard3), and persistent concerns regarding whether the method truly isolates safety features or introduces "real vs. synthetic" confounding variables.

**Reviewer Concerns:**

The authors successfully addressed requests for transparency by adding detailed human verification protocols and a dataset datasheet to the appendix. They also clarified misconceptions regarding the definition of "synthetic" images compared to prior work like UnsafeBench.

However, substantial concerns remain outstanding. Reviewers i781 and 4LMv argue that the benchmark lacks sufficient novelty and diversity because it is strictly derivative of LlavaGuard. More critically, the evaluation is considered incomplete due to the absence of comparisons with dedicated safety models (e.g., LlamaGuard3, ShieldAgent), which are standard baselines for this domain. Furthermore, despite author clarifications, skepticism persists regarding the "real vs. synthetic" distribution shift, with reviewers unconvinced that the reported performance gains reflect true out-of-distribution generalization rather than artifact learning.

**Reviewer Scores:**

Reviewers hn4i and 9SNW (both Score 6) reacted positively to the rebuttal and would likely maintain their scores, valuing the practical utility of the data pipeline. Conversely, Reviewers i781 and 4LMv (both Score 2) are expected to maintain their rejection. Reviewer i781 explicitly noted after the rebuttal that their concerns about generalization remain, and the fundamental issue of missing baselines raised by 4LMv cannot be resolved without new experiments that were not provided.

---

### Decision · Program_Chairs · 2026-01-26

Reject